# Cryo-EM structure of the mature and infective Mayaro virus at 4.4 Å resolution reveals features of arthritogenic alphaviruses

Helder V. Ribeiro-Filho[1,3], Lais D. Coimbra[1,3], Alexandre Cassago[2], Rebeca P. F. Rocha[1], João Victor da Silva Guerra [1], Rafael de Felicio [1], Carolina Moretto Carnieli [1], Luiza Leme[1], Antonio Cláudio Padilha[2], Adriana F. Paes Leme [1], Daniela B. B. Trivella [1], Rodrigo Villares Portugal[2], Paulo Sérgio Lopes-de-Oliveira [1] & Rafael Elias Marques [1✉]

Mayaro virus (MAYV) is an emerging arbovirus of the Americas that may cause a debilitating arthritogenic disease. The biology of MAYV is not fully understood and largely inferred from related arthritogenic alphaviruses. Here, we present the structure of MAYV at 4.4 Å resolution, obtained from a preparation of mature, infective virions. MAYV presents typical alphavirus features and organization. Interactions between viral proteins that lead to particle formation are described together with a hydrophobic pocket formed between E1 and E2 spike proteins and conformational epitopes specific of MAYV. We also describe MAYV glycosylation residues in E1 and E2 that may affect MXRA8 host receptor binding, and a molecular "handshake" between MAYV spikes formed by N262 glycosylation in adjacent E2 proteins. The structure of MAYV is suggestive of structural and functional complexity among alphaviruses, which may be targeted for specificity or antiviral activity.

[1] Brazilian Biosciences National Laboratory, Brazilian Center for Research in Energy and Materials (CNPEM), Campinas, SP, Brazil. [2] Brazilian Nanotechnology National Laboratory, Brazilian Center for Research in Energy and Materials (CNPEM), Campinas, SP, Brazil. [3] These authors contributed equally: Helder V. Ribeiro-Filho, Lais D. Coimbra. ✉email: rafael.marques@lnbio.cnpem.br

Mayaro virus (MAYV) is an emerging arbovirus in Central and South America[1–3]. Similar to arboviral human pathogens such as Chikungunya virus (CHIKV), MAYV is transmitted by mosquitoes and usually kept in a sylvatic cycle between mosquitos and non-human primates. Human infection with MAYV may lead to Mayaro fever, a dengue-like febrile illness characterized by joint pain which may persist for months[4,5]. MAYV biology and ability to cause disease remains poorly understood and is largely inferred from related alphaviruses. In recent years, MAYV infections have increased and expanded towards urban areas in Brazil[6–8], indicating that MAYV could become the next arboviral epidemic. There is no treatment or vaccine available against MAYV.

MAYV is an enveloped single-stranded RNA *Alphavirus*[9]. Mature, infective alphaviruses are icosahedral enveloped particles of ~70 nm in diameter composed of the structural proteins capsid (C), E1 and E2. The nucleocapsid core contains a single copy of the RNA genome surrounded by copies of the C protein, which is contained within the lipid envelope. The E1 and E2 proteins are transmembrane proteins that are organized in heterodimers. Trimers of E1 and E2 heterodimers compose the spikes at the viral surface that extend through the envelope bilayer and interact with nucleocapsid C proteins[10–13]. Spikes are involved in the attachment to cellular receptors[14], cell internalization and membrane fusion. Release of MAYV RNA[1] in the cytoplasm results in the expression of viral proteins, viral replication and culminate in the generation of mature and infectious viral progeny[15,16].

A common feature between enveloped viruses is glycosylation, which is important for virus attachment and entry into the host cell[17,18], as well as an escape strategy from the host's immune response[19,20]. Viruses depend on the host glycosylation machinery for post-translational modification of the viral proteins, which may be *N*-, *O*- and *C*-linked. Previous studies have described *N*-glycan residues in E1 and E2 of both encephalitic and arthritogenic alphaviruses[1] and are associated to viral attachment to host cells[21] and pathogenicity in vivo[22]. Unfortunately, the visualization of viral glycosylation in cryo-electron microscopy (cryo-EM) structures is challenging, as carbohydrate chains are flexible and exposed to solvent. The first carbohydrate monomers, however, are less flexible and could be located in high resolution structures, such as in the cryo-EM structure of Eastern Equine Encephalitis virus (EEEV)[1].

Here we describe the structure of mature and infective MAYV particles obtained by cryo-EM at 4.4 Å resolution, allowing the observation of *N*-glycosylation sites in MAYV E1–E2 spikes. MAYV has typical alphavirus features and organization. Comparison of the structure of MAYV to that of CHIKV reveals particularities on the E1 and E2 proteins that may be strategically explored for both therapeutics and diagnosis. We observed that MAYV glycosylation may affect MXRA8 receptor binding and spike stability and characterized a hydrophobic pocket in the core of MAYV E1–E2 heterodimers. Altogether, we describe features of MAYV that contribute to a greater understanding of alphaviral structure and biology.

## Results and discussion

**Cryo-EM structure of mature MAYV.** The cryo-EM structure of MAYV IQT4235 strain, originally isolated from a symptomatic patient in the Peruvian Amazon, was obtained at a global resolution of 4.4 Å (Supplementary Fig. 1) from a preparation of mature and infectious MAYV propagated in Vero CCL81 cell cultures (PDB entry 7KO8). The infectiveness of the purified MAYV was verified using plaque assays (Supplementary Fig. 2). The presence of the E3 protein was not observed, nor in the cryo-

EM density map (Supplementary Fig. 3A), even with a low-density threshold value, nor in SDS-PAGE gel carried out with the purified MAYV used for cryo-EM data collection (Supplementary Fig. 3B). In Semiliki Forest Virus (SFV) replication, E3 proteins detach from E2 proteins at the viral surface during maturation, under neutral pH conditions, a mechanism also observed in MAYV maturation[15,23,24]. Thus, our data indicate that the MAYV particles used in our study were mature and infective.

The MAYV cryo-EM density map reveals a common structural organization shared within members of the *Alphavirus* genus. A total of 80 spikes, formed by trimers of E1–E2 heterodimers, protrude from the MAYV membrane (Fig. 1A and B). The spikes are organized along two, three and fivefold icosahedral axes and a quasi-threefold symmetry axis for the E1–E2 trimer (q3) can also be observed (Fig. 1A and B). Similar to other alphaviruses, the MAYV asymmetric unit (ASU) is composed by four E1–E2-capsid units (T = 4) (Fig. 1E) and maintained by eight main interaction interfaces. A complete list of contacts performed among the MAYV structural proteins is presented in Supplementary Fig. 4.

The central cross-section of the MAYV cryo-EM density map shows the internal architecture of the virus (Fig. 1C and D). E1 and E2 proteins immerse into the lipid membrane through their transmembrane domain (TM) and the E2 C-terminal connects the spike to the capsid. The core of the MAYV particle, occupied by single-stranded RNA viral genome, is disordered and not resolved in this MAYV structure, given the imposition of icosahedral symmetry.

Assessment of local resolution in the MAYV density map shows that the C protein core and basal regions of E1 and E2 have the better resolution values (Fig. 1A and C). In contrast, the most external portion of E2 protein have lower resolution in comparison to the rest of E1–E2 ectodomains, indicating its higher flexibility, as well as the terminal portion of the two glycosylation sites (to be described below). Interesting, MAYV protein domains located at twofold axis, such as TM helices and basal regions of E1 and E2, tends to be more flexible in comparison to the ones in five or threefold axis (Fig. 1C).

**MAYV E1 and E2 ectodomains.** The ectodomains of E1 and E2 in alphaviruses are outside the lipid membrane and comprise solvent-exposed subdomains. In alphaviruses, the E1 protein is composed by 3 subdomains: I, II and III, whereas the E2 subdomains are more spatially segregated, divided into 4 subdomains: A, B, C and D (Fig. 2A). E1 and E2 ectodomains interact with each other, maintaining the spike structural organization. Newly putative side chain contacts are evidenced in our cryo-EM density map (Fig. 2B and C).

At the base of ectodomains and initial portion of TM subdomains, E1 and E2 (D subdomain) interact mainly through the contact between W407 from E1 and H352 from E2. (Fig. 2C—panel 1). Close to this region, other two histidine residues (H384 and H348) interact with positively charged residues (the latter also contacts T403) (Fig. 2C—panel 2). Towards the most exposed envelope regions, multiple contacts are observed between the E1 I subdomain and the E2 C subdomain (Fig. 2C—panel 3). In this region, we observe the proximity of positively charged residues to phenylalanine residues in addition to a possible hydrogen bond between R297 from E2 and A255 or N252 from E1 that seems to stabilize this core of E1–E2 interface. At the outer portion of ectodomains (E1 II and E2 B subdomains), two close contact regions are clearly observed between E1 and E2. The first one corresponds to the E1 fusion loop that protrudes between E2 B and A subdomains (Fig. 2C—panel 5) and involves

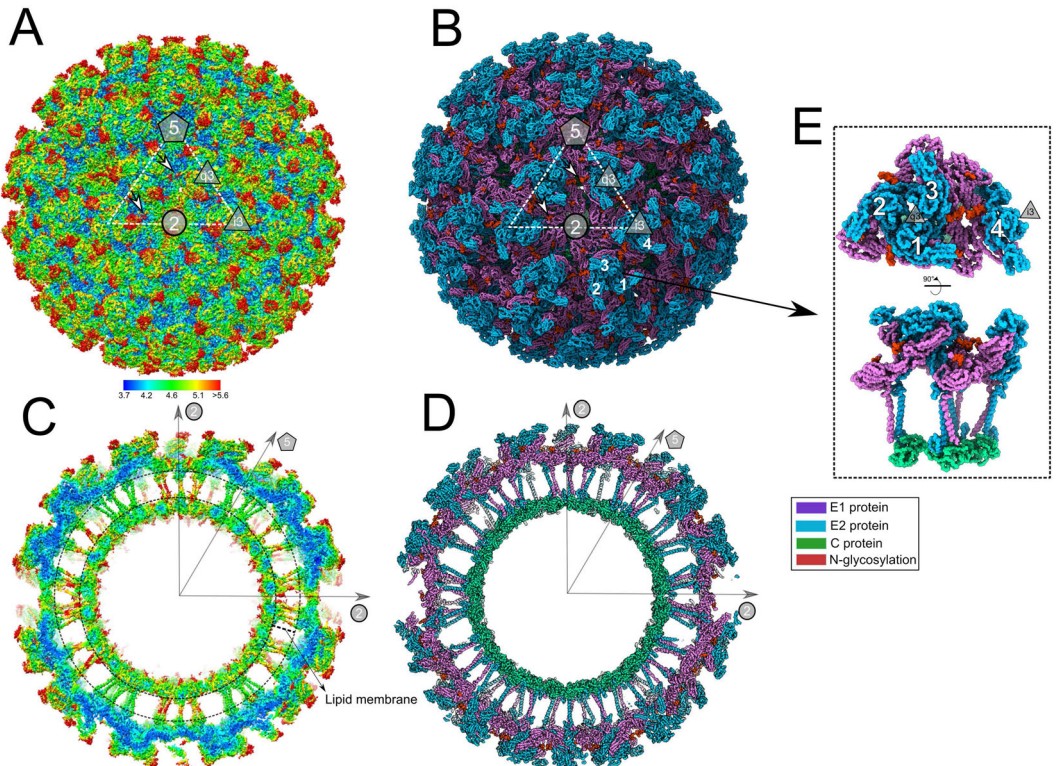

**Fig. 1 Overall structure of infectious MAYV determined by cryo-EM. A** MAYV particle showing the external surface with assigned symmetry axes. The white dashed triangle indicates the icosahedral asymmetric unit. Polygons indicate the fivefold axis (pentagon), three and quasi threefold axes (triangles), and twofold axis (circle). MAYV density map is in surface representation and coloured by local resolution estimated using ½-bit threshold criterion in Imagic-4D software system. Scale bar indicates the calculated local resolution. The local resolution values range from 3.7 to 7.8, with median of 4.2. **B** MAYV 3D atomic model fitted into the density map. For clarity, only backbone atoms are presented. The model shows coloured MAYV proteins. 5-3-2 symmetry axes and the asymmetric unit are the same as in (**A**). **C**, **D** Central cross-section of the MAYV density map or atomic model, respectively. **E** Two rotated views of MAYV asymmetric unit formed by four E1–E2 heterodimers following icosahedral *T* = 4 triangulation. Three and quasi threefold axes are indicated.

W89 from E1 and H73 and H29 from E2 (Fig. 2C—panel 5). A continuous density is also observed between residues N228 and Y18, which seems to contribute to fusion loop stabilization. The second close contact comprises a putative hydrogen bond between S57 from E1 II subdomain and N238 from E2 A subdomain (Fig. 2C—panel 4) in addition to the interaction between R244 and Y59. The former residue pair is highly conserved in alphaviruses (Supplementary Fig. 5) and together with the fusion loop can contribute to the stabilization of the outer portion of E1–E2 interface.

Other contacts involving different heterodimers are also evidenced by the density map. A basic residues rich loop from E2 C subdomain fits between E1–E2 outer subdomains from the neighbouring heterodimer (Fig. 2C—panel 6). Contacts between sidechains of E1 ectodomains from different spikes are also observed: the E1 N-terminal is potentially stabilized by interaction between Y1 and K383 from another E1 and a symmetric contact between two residues H152 and Y192.

The high frequency of histidine residues forming the E1–E2 interface reflect the plasticity of E1–E2 pH-dependent interaction and their role in the spike assembly/disassembly process. At low pH, such as in endosome, histidine residues are protonated and positively charged. Close contacts with basic residues (Fig. 2C— panel 2), can become repulsive, as well as contacts with aromatic residues in a hydrophobic environment (Fig. 2C—panel 1, 5, 6 and 8), could be non-favourable. During the virus lifecycle, the reduction in E1–E2 affinity yields E1 homotrimer and the fusion complex.

**Comparison of MAYV and CHIKV ectodomains and neutralizing antibody sites**. Given the relevance of E1–E2 ectodomains in host cell infection and induction of humoral immune responses, we compared the MAYV structure to CHIKV. These alphaviruses are closely related, cause similar diseases, and induce serological cross-reactivity that complicates diagnosis[25]. In comparison to CHIKV, MAYV has three missing residues. The first two are sequential and located in the beta-sheet from the E1 III subdomain, at the base of the E1 ectodomain, exposed to solvent but close to the lipid membrane. These missing residues do not disrupt the overall tertiary E1 III subdomain structure, but result in a shorter beta-strand, as observed in Fig. 3A (Right Inset, purple ribbon). Consequently, the number of contacts between two antiparallel beta-strands reduce, increasing the distance between them. Conformational epitopes present in this region would be changed and could be used in the rational design of antibodies to differentiate MAYV from CHIKV. The third missing residue is an asparagine in the E2 ectodomain, at the loop between A and B E2 subdomains (Fig. 3A, left inset). Additional asparagine residues could create new glycosylation sites, but in this case, the CHIKV asparagine is not in a canonical glycosylation motif and there is no evidence of glycosylation in that site, as checked in the CHIKV cryo-EM structure (PDB 6NK5)[26].

There are several non-conserved residues on the surface of MAYV E1 and E2 ectodomains in comparison to CHIKV (Fig. 3B and C). The exposed surfaces of alphavirus ectodomains are targets for neutralizing antibodies, which were mapped in MAYV using a putative neutralizing antibody binding zone explored in

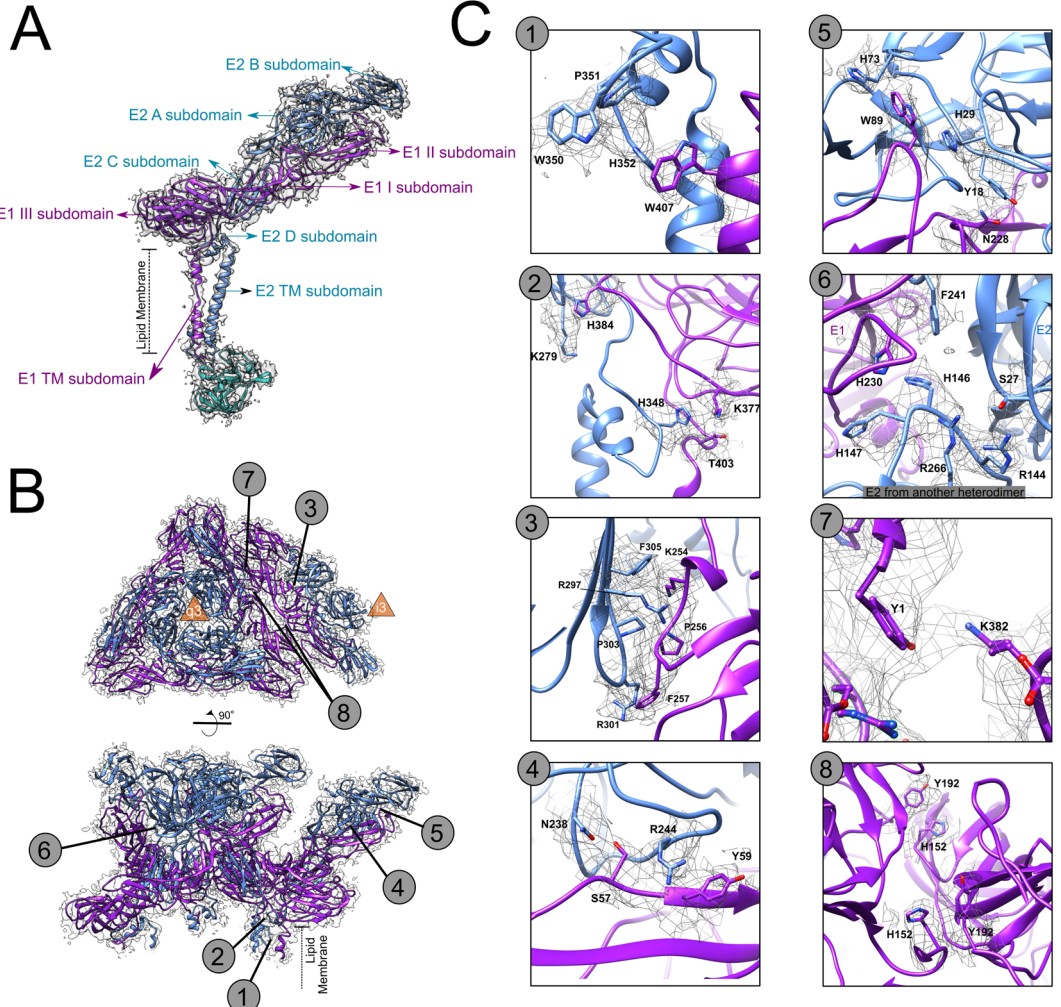

**Fig. 2 Organization of MAYV E1 and E2 ectodomains. A** Overall organization of MAYV subdomains from E1 and E2 heterodimers. **B** Upper view and side view of E1 (purple) and E2 (blue) ectodomains from an ASU. The quasi-3-fold axis is indicated at the center of the trimeric spike. Numbers indicate the corresponding zoomed images in (**C**). **C** Zoomed in images of MAYV ectodomains highlighting structural sidechain contacts involved in E1–E2 function and organization. MAYV density map is in mesh representation.

CHIKV (Fig. 3B). This zone comprises mainly the E2 A and B subdomains and have several non-conserved residues between MAYV and CHIKV (Fig. 3C). Some MAYV-neutralizing antibodies cross-react with other alphaviruses (CHIKV, UNAV, RRV, ONNV) and preferentially target three epitopes of E2 B domain: residues 181–190; 205–211 and 214–218[5]. We noticed positively charged residues that are not included in the cross-reacting epitopes and are not conserved in MAYV (MAYV K58, K234, R195) (CHIKV K57, K233, Q195). These amino acid sequences could compose potential epitopes for virus-specific antibodies. We suggest these differences should be explored in the rational design of antibody-based diagnostic methods to identify infections using MAYV exclusive epitopes.

**MAYV N-glycosylation sites and the "handshake" configuration.** MAYV has two predicted N-glycosylation sites: N141 in E1 and N262 in E2. We observed both glycosylation sites in the MAYV cryo-EM density map as protruding densities at asparagine side chains (Fig. 4A–C). We also assessed predicted O-glycosylation sites in MAYV E proteins. Only one predicted O-glycosylation site was found: S238 within the sequence TPSGF. However, the density map around this residue did not support glycosylation. To check whether there were other visible

glycosylation sites in the MAYV cryo-EM structure, we manually inspected for extra densities near to all exposed serine and threonine residues, but no other density was found.

The N262 N-glycosylation site is conserved in MAYV, CHIKV, SFV and RRV, but absent in Sindbis virus (SINV) and EEEV, either in the linear sequence or spatially. In SINV and EEEV, the asparagine residue in E2 is substituted by threonine and no equivalent N-glycosylation sites were found in the corresponding E2 proteins. N-Glycosylation at MAYV E2 N262 protrudes in a long, continuous and branched density in direction to a symmetric N-glycosylation at the neighbouring E2, which is reminiscent of a molecular handshake (Fig. 4A). The straight proximal portion of the density can accommodate three carbohydrates, which were modelled as canonical NAG-NAG-BMA (NAG: N-acetyl-β-D-glucsamine and BMA: β-D-mannose). At the position of the third carbohydrate, a branched density can be observed, in which 1-3MAN (α-D-mannose) and 1-6MAN monomers were modelled. Additional densities can be observed beyond this branch, but due to the weak density, no other carbohydrate was modelled. The straight disposition of the N262 E2 N-glycosylation density can be supported by contacts with protein residues. The proximity between E1 H116 and the first NAG may produce a putative hydrogen bond (Fig. 4B, left) and

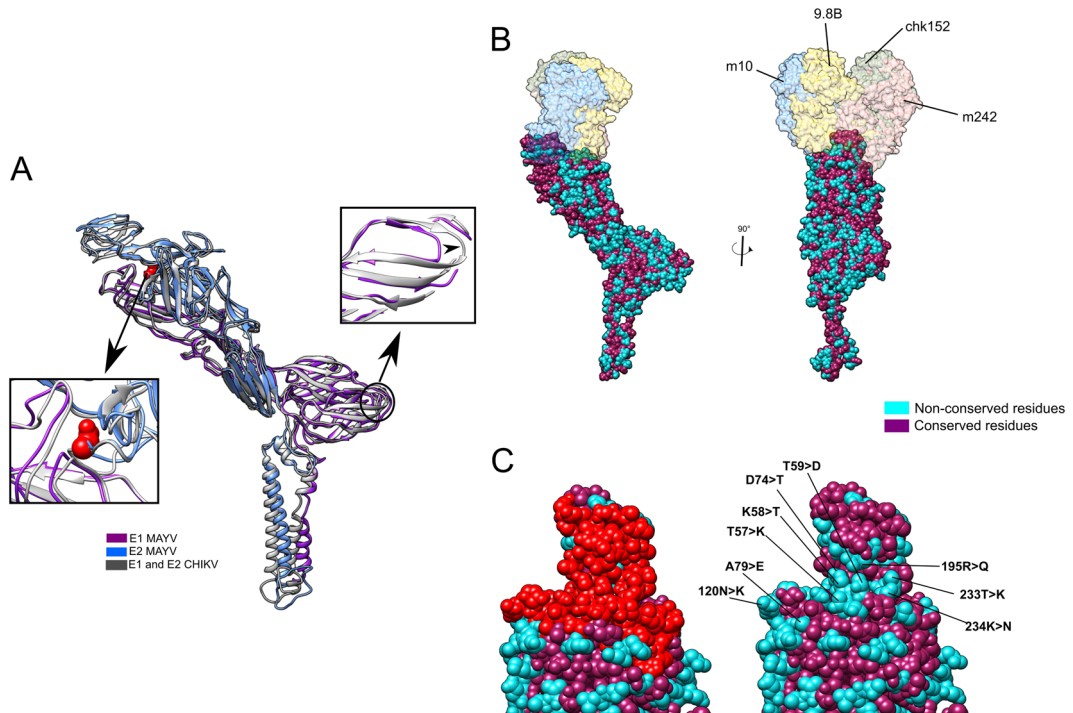

**Fig. 3 Comparison of E1–E2 ectodomains structure and sequence from MAYV and CHIKV. A** Structural alignment of E1 and E2 domains from MAYV (blue and purple) and CHIKV (grey). The shorter MAYV loop in the E1 III subdomain is highlighted in the right inset. Details of the loop connecting the E2 A and B subdomains, including the missing asparagine in MAYV, are shown in the left inset. **B** E1–E2 dimer coloured by conservation in relation to CHIKV and aligned with CHIKV structure in complex with Fab fragments from CHIKV-neutralizing antibodies CHK-9, CHK-152, m242 and m1040. **C** The interaction sites for the four antibodies are highlighted in red in the MAYV structure. The conservation map on the antibody interaction surface is highlighted, with non-conserved residues in MAYV labelled in cyan.

the change of histidine protonation state upon environment pH variation could affect glycan interaction and alter glycan stability. The handshake configuration of the E2 N262 *N*-glycosylation was not observed in other alphaviruses despite the glycosylation site being present in CHIKV (N141, entry 6NK5)[21].

The N141 glycosylation site has a less defined density in comparison to N262 (Fig. 4B), as only the canonical NAG-NAG-BMA glycan sequence could be modelled. The weak density may indicate a more flexible *N*-glycosylation site. N141 is conserved in MAYV, CHIKV, SFV and RRV but not in other alphaviruses, such as SINDV, EEEV and VEEV. Despite not being located at the same position of N141, SINDV has a glycosylated asparagine (N139) two residues upstream the MAYV N141, whereas EEEV and VEEV have another site (N134) six residues upstream. The EEEV and VEEV glycosylation sites are distant from MAYV N141 in the linear sequence alignment (Supplementary Fig. 4), but the beta-sheet protein folding brings the EEEV and VEEV glycosylation sites spatially close to MAYV N141 (Fig. 4C). This finding indicates a structurally conserved *N*-glycosylation site in alphaviruses.

The glycosylation sites were confirmed by LC-MS/MS. For glycosylation site characterization, three complementary strategies were used: (i) only trypsin digestion, (ii) complete release of N-glycans through PNGase F digestion followed by trypsin digestion, or (iii) Endo H digestion, which cleaves oligomannose and some hybrid N-glycans within the chitobiose core, leaving one N-acetylglucosamine residue on the asparagine, following trypsin digestion. With the first strategy, proteins E1 and E2 were identified with 69.04% and 78.20% of sequence coverage, respectively. In this approach, a peptide from E2 protein, TINSCTVDKCQAYVTSHTK (*m/z* 738.6825, +3) was the unique peptide that contains a possible motif of glycosylation showing

deamidation in N218. With the second strategy, the complete release of N-glycans by PNGase F results in deamidation of the asparagine previously N-glycosylated. The peptides VTYGTVNQTVEAYVNGDHAVTIAGTK (*m/z* 904.1144, +3) and with a missed cleavage IKVTYGTVNQTVEAYVNGDHAV-TIAGTK (*m/z* 738.6342, +4) from E1 protein were identified with deamidation in the N141. From E2 protein, the peptide VHIPFPLINTTCR (*m/z* 523.6142, 3+) was identified with a deamidation in the N262. Finally, considering that spontaneous deamidation can occur in asparagine[27], we used the third strategy with Endo H and trypsin digestions to verify if the possible glycosylation sites were occupied by oligomannose or hybrid N-glycans. This resulted in the identification of VTYGTVNQT-VEAYVNGDHAVTIAGTK (*m/z* 971.1465, +3) E1 peptide and VHIPFPLINTTCR (*m/z* 590.9796, +3) E2 peptide with the N-acetylglucosamine residue on the asparagine (Supplementary Fig. 11). Complete lists of protein groups and peptides identified in each digestion condition are presented in Supplementary data files 1–6. Therefore, combining the different strategies, the results indicate the glycosylation in N141 of E1 protein and in N262 of E2 protein.

The general composition of E1/E2 glycans could be assigned based on UPLC-MS/MS analyses of the released glycans derived from E1/E2 glycoproteins digested with PNGAse F and derivatized with procainamide. Nine peaks relative to E1/E2 glycans could be observed, with 7–11 carbohydrates represented by combinations of N-acetyl-hexosamines (HexNAc) and hexoses (Hex)—(Fig. 4D). HexNAc is pointed as the first monomer in the glycan based on its identification as a complex with procainamide in the MS[2] data (*m/z* 441.2700, +1). The combination of two HexNAc and hexoses are represented in the predominant peaks 3, 8 and 9 (Fig. 4D and Supplementary Table 5) and are suggestive of oligomannose

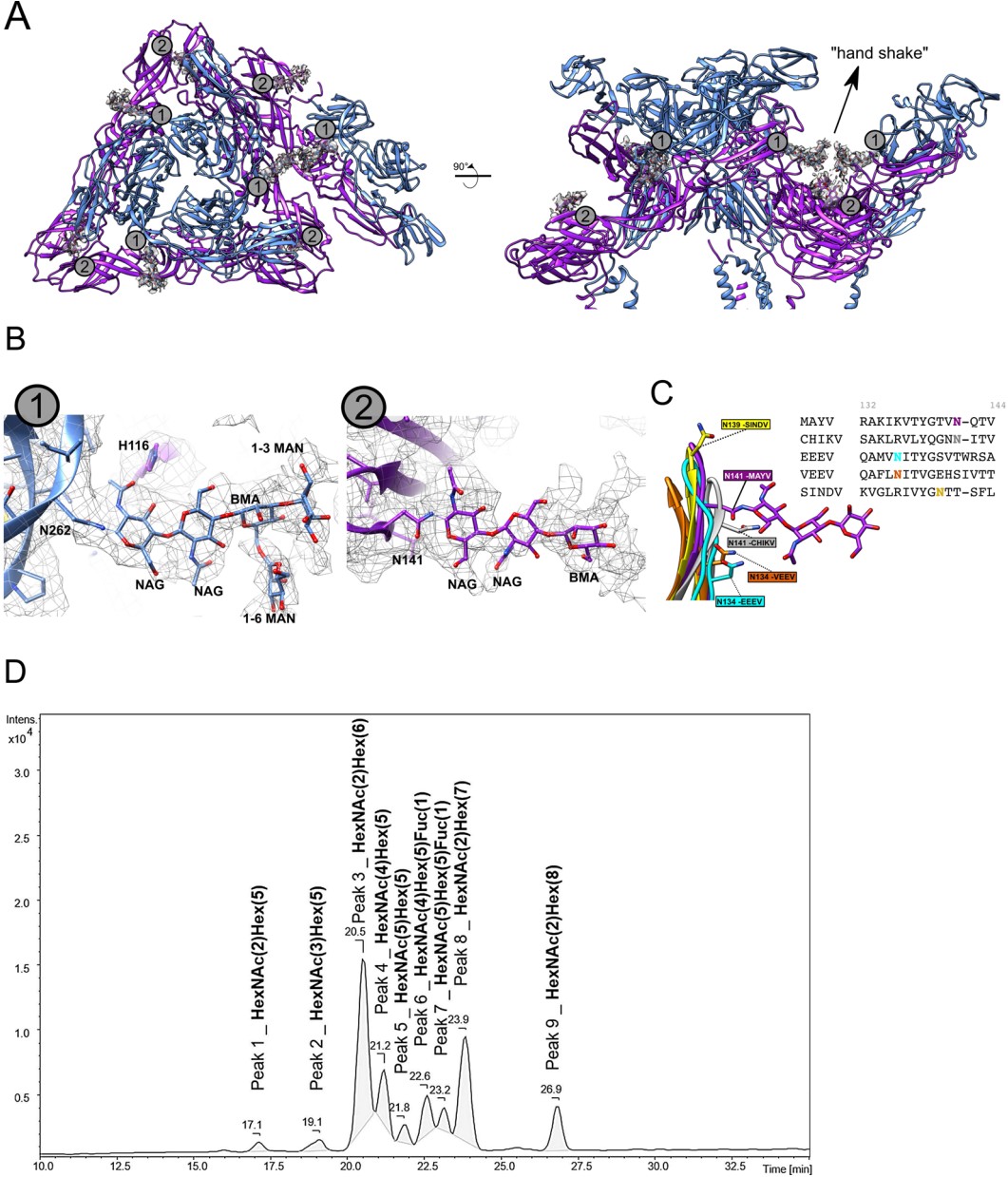

**Fig. 4 N-Glycosylation sites observed in the E1–E2 ectodomain of the MAYV cryo-EM structure. A** E1 (N141) and E2 (N262) N-glycosylation sites. **B** On the left, zoom at the E2 N262 glycosylation with the carbohydrate monomers modelled. Polar residues under hydrogen bond distances are shown in sticks and labelled. On the right, detail of the E1 N141 site with the carbohydrates modelled. **C** Equivalent asparagine residues regions found in EEEV (cyan), VEEV (orange), SINDV (yellow) and CHIKV (grey) cryo-EM structures are shown superposed to the MAYV structure. MAYV density map is in mesh representation. **D** Base peak chromatogram derived from UPLC-MS/MS analyses of N-glycans released from E1/E2 glycoproteins of MAYV. Nine peaks were integrated (#1 to #9), being further characterized based on the MS1 isotopic and MS2 fragmentation patterns as N-glycans composed by combinations of N-acetyl-hexosamine (HexNAc); hexoses (Hex) or fucose (Fuc) as indicated. GlcNAc (NAG) and Man (MAN) were assigned as the hexosamine and hexose monomers based on enzyme PNGase F and EndoH specificity and cryo-EM density map interpretation.

N-glycans. This finding corroborates the NAG-BMA-MAN core fitted into the Cryo-EM map at position N262. The other peaks, with more than two HexNAc and containing fucose are possibly hybrid or complex N-glycans. Overall, these data also corroborate the LC-MS/MS analyses of E1/E2 protein digests with PNGase F and EndoH presented above. From these orthogonal data (enzyme specificity and cryoEM density map), and in line with the cannonical glycosylation residues expected in mammalian cells, NAG and MAN can be suggested as the N-acetyl hexosamine and hexose carbohydrates of MAYV glycans detected in the UPLC-MS/MS analyses. In addition, glycans eluted as peaks #6 and #7 further presented one fucose (Fuc) monomer in their composition, with a fragment peak of m/z 587.3284 (+1), corresponding to the fucose being attached to the procainamide derivatized HexNAc. We did not observe densities compatible with fucose linkage to the NAG core in the Cryo-EM map, which may suggest low prevalence of such linkage or fucosylation at distal NAG. Please refer to Supplementary Fig. 12, Supplementary Table 5 and data deposited at GlycoPOST (GPST000171) for further details on UPLC-MS/MS data acquisition and interpretation.

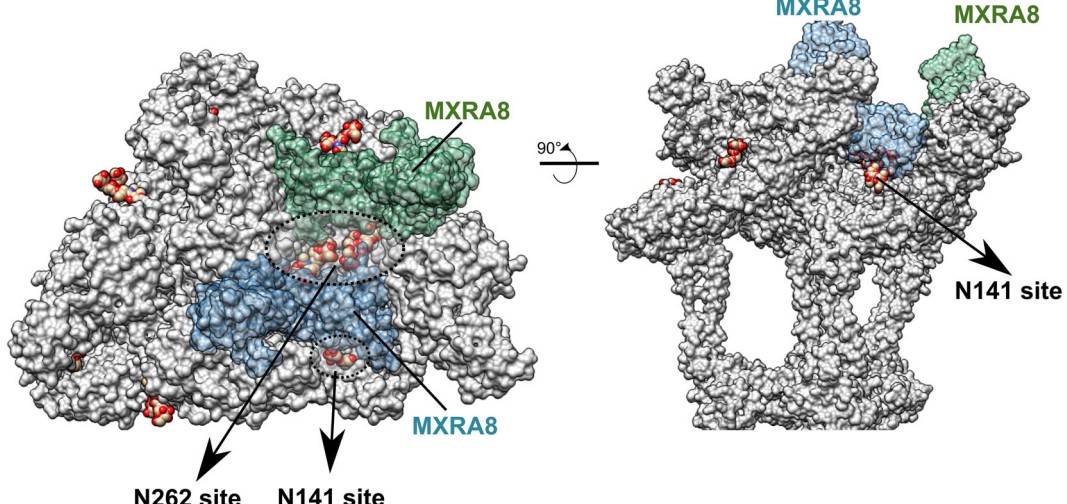

**Fig. 5 MAYV glycosylation sites may play a role in the interaction with the MXRA8 receptor.** The E1 and E2 monomers are shown as grey surface. The MXRA8 receptor surfaces were mapped in the MAYV cryo-EM structure by superposing the cryo-EM structure of murine MXRA8 in complex with CHIKV (PDBid 6NK624). The N141 (E1) and N262 (E2) glycosylation sites are highlighted by dashed lines and pointed. Top (left) and side (right) views are shown. The N262 handshake glycosylation site is located at the center of the MXRA8 interaction site and potentially participates in the interaction with the host MXRA8 receptor.

The region occupied by the glycans may be important for receptor binding, especially the N262 site. MXRA8 is a well-described host receptor that has an important role in alphavirus infection[14] and its cryo-EM structure in complex with CHIKV has been reported[26]. By superposing the CHIKV cryo-EM structure with the MAYV structure it is possible to observe that the binding site of the murine MXRA8 is close to the MAYV N262 *N*-glycosylation site (Fig. 5). Considering a similar MXRA8 binding site in CHIKV and MAYV, the N262 glycosylation would be adjacent[28]. The infection by alphaviruses that do not have the N262 glycosylation site (SINDV, VEEV and EEEV) is not MXRA8 dependent[14], suggesting that the glycans may participate in MXRA8 binding.

**E1 and E2 transmembrane helices and the hydrophobic pocket.** The E1 and E2 TM helices are mostly composed by hydrophobic residues. Despite not entirely conserved in other alphaviruses, substitutions in the TM helices are chemically conserved (replacement among aliphatic residues) (Supplementary Fig. 5). Surrounding the hydrophobic residues, especially in the middle of the E2 TM helix, serine residues occupy the center of the lipid bilayer in a zone with weaker density (Supplementary Fig. 6). Polar residues in helices can contribute to TM packing and bending[29]. Glycine residues in E1 also interrupt the sequence of hydrophobic side chains and break the helix. At least two out of the three serine residues in E2 TM are conserved in CHIKV and SFV. In E1 TM, the three glycine residues are conserved among alphaviruses (Supplementary Fig. 5), indicating the importance of these serine and glycine residues for the transmembrane structure of alphaviruses.

The core region between E1 and E2 forms a cavity that is occupied by a long extra density, which cannot be explained by side-chain residues (Fig. 6A and B). A similar density profile was previously observed in a SINV cryo-EM map and the authors hypothesized that a hydrophobic phospholipid tail (C18), named as "pocket factor", could occupy that density and stabilize the hydrophobic pocket formed between E1 and E2[30]. The MAYV extra density is long enough to fit a C18 hydrocarbon lipid molecule (Supplementary Fig. 7), displaying a similar size in comparison to SINV. The continuous density between the small

molecule and the Y358 aromatic group indicates a close hydrophobic contact that seems to stabilize the ligand. Y358 is highly conserved in alphaviruses (Supplementary Fig. 5) and suggests a key role for this tyrosine in alphavirus structure. To further understand the pocket environment and extract its chemical features, the whole MAYV pocket was prospected using the parKVFinder software[31] and compared to other alphaviruses. In MAYV, the pocket between E1 and E2 TM domains has a volume around 850 Å$^3$ (Supplementary Fig. 8). This volume is quite similar in SIND, EEEV and VEEV, but in CHIKV E1 and E2 are more distant from each other, which creates a larger volume. The hydrophobic nature of the pocket in alphaviruses is clearly noticed by the hydrophobicity surface mapping (Fig. 6C) and by the number of apolar residues forming the cavity core (Supplementary Fig. 7). The pocket density extends to polar residues, such as H362, T365 and T366 (from E2) at the posterior opening (Supplementary Fig. 7), indicating that the molecule could have an amphipathic nature, as a fatty acid. T365 and T366 have a conserved structural position in other alphaviruses or are changed to serine, an even more polar residue. At the posterior opening, another histidine (H352 from E2) helps to close the pocket. Most alphaviruses, except SINV, have histidine residues occupying a similar position. Altogether, these findings indicate that alphaviruses have a consistent amphipathic cavity formed between E1 and E2 domains at the outer leaflet membrane. Should the alphaviral pocket be occupied by a molecule, such molecule would be chemically similar in different alphaviruses. The MAYV density map suggests the extra density could be occupied by a fatty acid that may enhance interactions between E1 and E2. Thus, the pocket could be targeted for the development of antiviral compounds against MAYV and other alphaviruses, using rational drug design.

**MAYV C-E2 interactions.** C proteins from alphaviruses are formed by two subdomains, an N-terminal disordered domain that binds to viral RNA, which is not observed in MAYV density map, and a C-terminal structured domain that attaches to the E2 proteins. The N-terminal region has lower identity within alphaviruses (Supplementary Table 1) and is reported as virus-specific[32]. In contrast, sequence identity of the structured portion

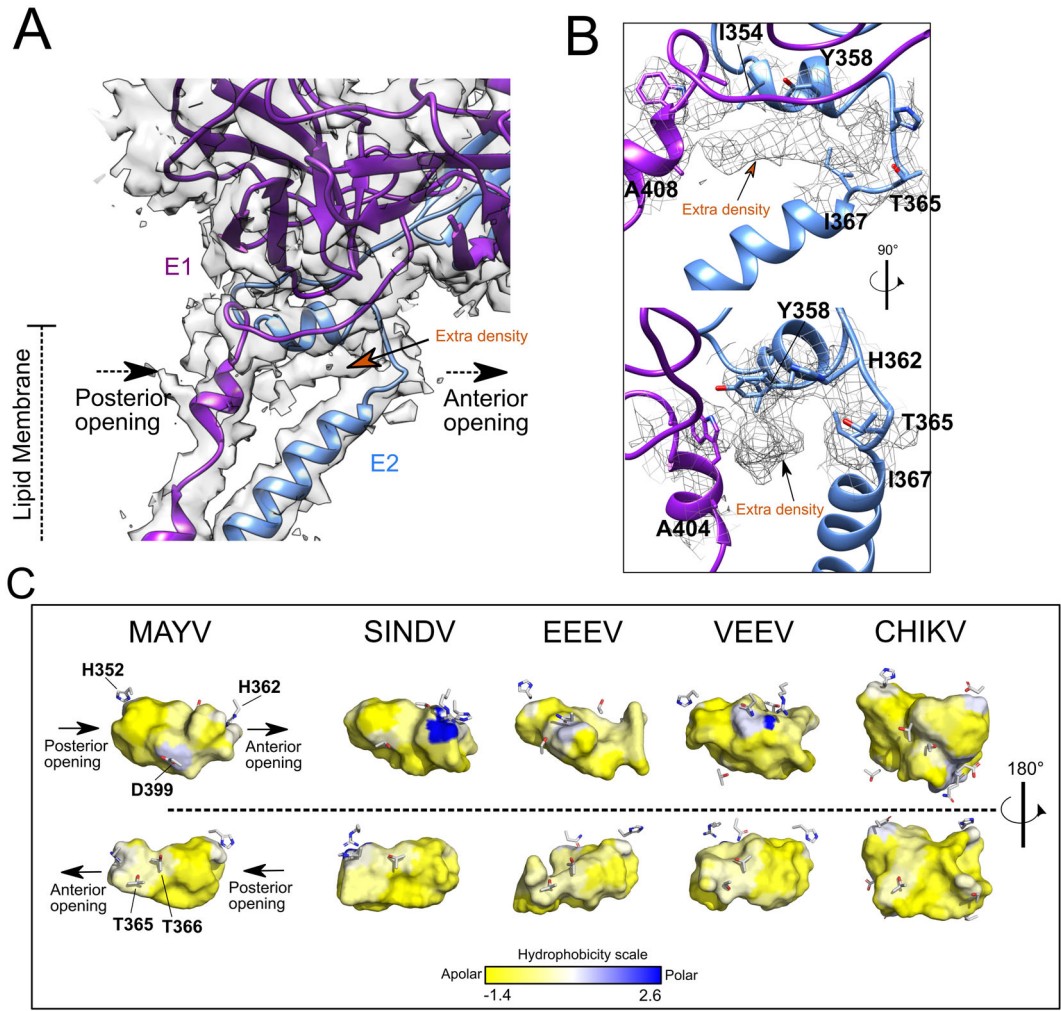

**Fig. 6 MAYV E1 and E2 transmembrane domains and the hydrophobic pocket. A** MAYV 3D atomic model fitted into the density map showing the top portion of E1 and E2 TM helices. E1–E2 intersection forms a cavity with an anterior and posterior opening. The extra density observed inside the cavity is indicated. **B** Detail of the extra density found at the E1–E2 head and residues surrounding it. MAYV density map is in surface or mesh representation. **C** Cavity prospection in alphaviruses structures. The cavity between E1 and E2 domains is in surface representation and is coloured based on the Eisenberg consensus scale[63] and using the residues forming the cavity, as detailed in the methods section. Only polar residues are represented as sticks.

of C protein is ~90% between MAYV and CHIKV, SFV or RRV, leading to a similar C-protein folding (Supplementary Table 1). The ordered portion of alphaviral C proteins is pivotal for particle organization and assembly, arranging the icosahedral nucleo-capsid and connecting to E2 and E1, in the envelope[32,33].

MAYV density map corroborates the conserved C-protein overall folding, forming two β-sheet rich subdomains separated by a shallow cavity (~500 Å$^3$, Supplementary Fig. 9) in which E2 C-terminal is non-covalently attached (Fig. 7A and D). The pocket bottom is hydrophobic whereas its ceiling has polar and charged residues (Fig. 7A and D). The capsid-E2 interface involves the well-described TPY consensus motif[34], which is conserved within the *Alphavirus* genus. The proline from this motif (P398) ensures the helix break and places the E2 C-terminal loop inside the C pocket (Fig. 7B). Our density map also indicates that the tyrosine in the TPY motif (Y399) points towards the hydrophobic residues V407 and P408 from the E2 loop, which might help to maintain the E2 C-terminal turn. A second proline (P403) two residues downstream the TPY motif potentially plays a key role in the capsid-E2 interaction (Fig. 7C). This proline is buried in the C protein pocket and a continuous density reveals a hydrophobic stacking with Y175[32,35]. Another aromatic residue (W242) is close to P403 and its neighbouring residue T402[36,37].

In CHIKV, the tyrosine residue Y175 is changed by phenylala-nine, maintaining the aromatic characteristic. Interestingly, small molecules proposed to inhibit the capsid-E2 interaction contain heterocyclic rings, which reinforces the relevance of this kind of contact for capsid-E2 interaction and highlight this site as a putative drug target[38].

**Capsid assembly is driven by electrostatic interactions.** The organization of alphavirus capsid was first described for RRV in 1995[35] when key aspects of assembly and maturation were defined. Our cryo-EM map suggests that MAYV capsid assembly is based on electrostatic contacts between the C proteins. The C proteins are organized as a layer of hexameric, located at twofold axis, or pentameric units, at fivefold axis (Fig. 8A). Despite the interface between C-proteins from the assembly units being the same, the C-protein core of the former presents more flexibility than the latter (Fig. 8A). The interface between C-protein from the same hexameric or pentameric assembly units is formed by amino acid residues of complementary charges. Contacts may involve symmetric electrostatic interactions between residues K169 from a capsid unit and E231 from another (Fig. 8B). E231 is conserved in the *Alphavirus* genus while K169 is substituted by arginine in SINV, VEEV and SFV, while maintaining the same

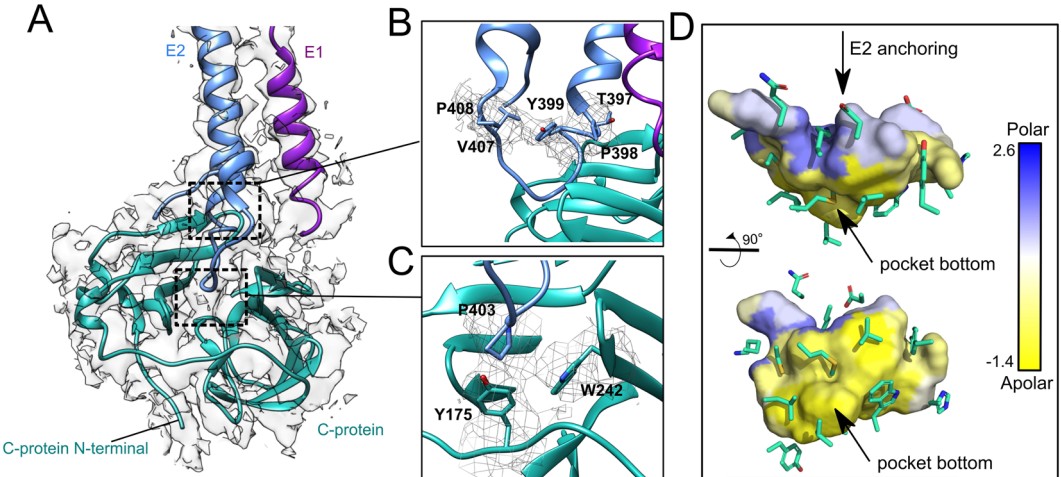

**Fig. 7 MAYV capsid interaction with E2 C-terminal domain. A** MAYV 3D atomic model fitted into the cryo-EM density map. C-, E1 and E2 proteins are represented as cyan, purple, and blue cartoon, respectively. **B** The TPY motif (residues T387, P398 and Y399) interaction with the C protein. **C** P403 and T402 from E2 and their interaction with aromatic residues Y175 and W242 in C protein. MAYV density map is shown in mesh representation. **D** Cavity prospection in MAYV C-protein showing a hydrophobic environment in the pocket bottom and a polar and charged environment at cavity outer edges. C-protein cavity that binds E2 C-terminal is in surface representation and is coloured based on the Eisenberg consensus hydrophobic scale. C-protein residues surrounding the cavity are represented as sticks.

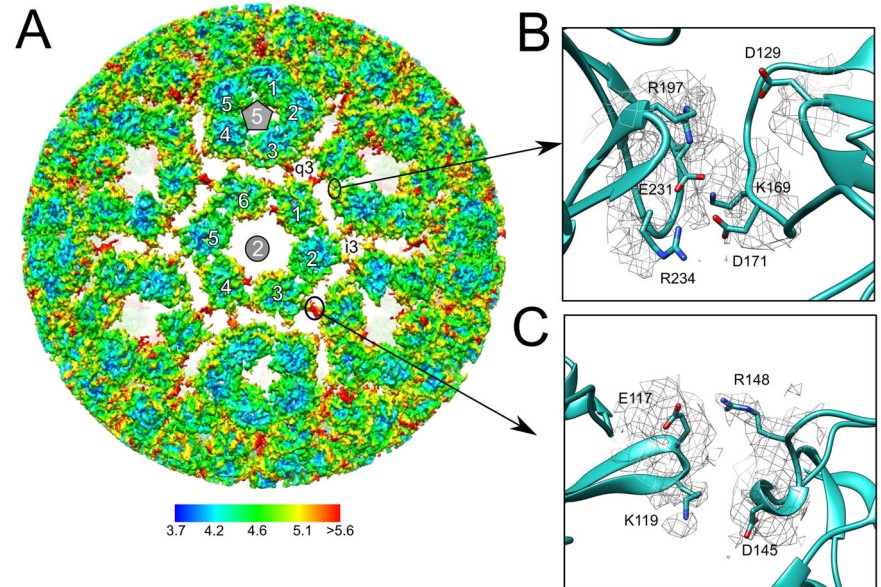

**Fig. 8 Organization of the MAYV capsid. A** MAYV density map showing the C-protein layer of hexameric and pentameric units coloured by local resolution estimated at a conservative FSC 0.5 threshold. The inset shows the putative electrostatic interactions between the C proteins. **B** Potential contacts between C proteins from the same hexameric unit and **C** between C protein from different assembly units.

chemical characteristic. A similar pattern was observed in SINV and CHIKV[39,40]. A second charge driven pair can also be observed for residues D171 and R234. Alike the former, this interaction pair is conserved in alphaviruses, except for the substitution of D171 for a glutamic acid residue in SINV, which again maintains its chemical properties. The density at the side-chain of these residues is weak, and we cannot define an ion pair or hydrogen bond formation. However, the conservation of these electrostatic pairs in C-protein interface indicates their relevance for the virus internal structure.

MAYV density map show that, in addition to the intrapenta-meric or intrahexameric interface described above, C-proteins from different hexameric or pentameric units became closer around 3 or quasi threefold axes and contact each other (Fig. 8A and C). Interestingly, we also found charged amino acid pairs in this interface (Fig. 8C) and the density at their sidechains is also not well defined, supporting a flexible interface. Even before viral budding, the presence of such charged pairs, in addition to the aforementioned interactions, may provide electrostatic forces that orient the association of C protein hexamers and pentamers during viral assembly. Electrostatic forces seem to be important for capsid organization in oligomeric units and for capsid spatial orientation relative to the lipid membrane[41]. This observation is supported by the electrostatic potential of the MAYV capsid surface (Supplementary Fig. 10). The top view of the MAYV capsid hexamer shows a highly positively charged region (Supplementary Fig. 10) formed by lysine residues, such as 130, 152, 153, 156 and 244 in MAYV, that are facing the lipid

membrane and guide the attachment of the C proteins to the negatively charged phospholipid heads. Most of these lysine residues, mainly K130, K153 and K244, are conserved in the *Alphavirus* genus.

We presented the MAYV cryo-EM structure obtained from mature and infective viral stocks. We reached a global 4.4 Å resolution and were able to observe structural features in the MAYV virion. Our findings expand the current understanding of MAYV biology, supports the development of potential treatments and specific diagnostic methods for an emerging alphavirus.

The 3D atomic model allowed the comparison of MAYV and its closely related alphavirus, CHIKV and state structural differences. A non-conserved cluster of residues was found in the putative neutralizing antibody (CHK-9, CHK-152, m242, and m10) epitopes in MAYV in comparison to CHIKV. Our findings confirm that there are specific epitopes between the closely related arthritogenic alphaviruses and provide a molecular basis for which neutralizing antibodies against CHIKV may be broadacting or not. Moreover, the rational design of MAYV antigens or antibodies that discriminate MAYV and CHIKV are of major relevance in Latin American countries that are endemic for both arboviral diseases. Cross-reactivity in convalescent human sera has contributed to sub notification of Mayaro fever cases in Brazil[42]. A correct assessment and diagnosis of MAYV infections are key to better understand disease epidemiology and formulate effective public health policies. These considerations could also facilitate the design of specific serological tests for SFV and RRV, which may also be misdiagnosed as CHIKV infections in other continents.

Key contacts between E1 and E2 ectodomains were revealed in the MAYV density map. Absence of E3 in the MAYV particle was evident in cryo-EM density map, protein electrophoresis and infectivity assays. Most subdomain contacts involve histidine residues, which have different protonation states varying according to the cell environment pH. At neutral pH, such as in the endoplasmic reticulum, these histidine residues would promote contacts that stabilize the interaction between E1 and E2 and thus, would play a role in virus assembly at the final stages of the MAYV replication cycle. In contrast, these histidine residues may also play a role in the first stages of virus infection, as MAYV must escape acidic endosomes and release the viral RNA into the cytoplasm. An acidic environment would cause histidine residues to change protonation states and facilitate the conformational changes that expose the fusion loop in E1 and lead to the fusion of viral and host membranes, thereby releasing viral RNA. In summary, histidine residues may be important to allow MAYV proteins to be dynamic and responsive to changes in pH involved in different steps in the replication cycle.

The careful inspection of the MAYV glycoproteins allowed the identification of two *N*-glycosylation sites that might be relevant for virus biology. The N141 in E1 was already reported as important for alphavirus infection of host cells[22]. Here we report the N262 stable *N*-glycosylation interface ("handshake" glycosylation site), that connects symmetric E2 proteins from adjacent MAYV spikes. This interaction might contribute to the stabilization of MAYV spikes and might modulate interaction with the MXRA8 mammalian receptor. This *N*-glycosylation site is not conserved among the *Alphavirus* genus, being present only in arthritogenic alphaviruses such as CHIKV and MAYV, suggesting that N262 glycosylation may be involved in virus tropism and in the pathogenesis of arthritogenic disease. Functional studies are required to assess the contribution of N262 handshake to MAYV particle stability and infectivity, which have direct implications for the design of antiviral strategies and the development of attenuated viruses for immunization.

The hydrophobic pocket located between E1 and E2, first described by Chen and colleagues[30], was also observed in the MAYV structure. Pocket analysis using parKVFinder[31] supports the hypothesis that a fatty acid occupies the region. Further structural and functional studies are necessary to confirm the composition of the "pocket factor", the function of the hydrophobic pocket in alphavirus biology and whether it could be targeted in the development of potential treatments.

Finally, conserved electrostatic pairs between C protein hexamers or pentamers, and conserved positively charged residues in C that may guide nucleocapsid assembly and interaction with the host lipid membrane. These findings raise the relevance of charged residues in MAYV C proteins and suggest that C proteins are central in alphavirus assembly. We could not visualize the N-terminal subdomain of C proteins or the MAYV genome, which remain unresolved. The C protein remains a promising target for the development of antiviral therapies against alphaviruses. Blockade of C serine protease activity, binding of C to viral RNA or anchoring of the nucleocapsid to E2 will certainly result in the impairment of viral replication.

## Methods

**Virus.** The MAYV strain IQT 4235 (GenBank accession number MK070491.1) originally isolated from a symptomatic patient in the Peruvian Amazon was kindly provided by Dr. Maurício L. Nogueira (FAMERP, Brazil). Vero CCL81 cells were purchased from Banco de Células do Rio de Janeiro (BCRJ, Brazil) and were confirmed to be mycoplasma-free.

**Preparation of viral stocks and MAYV purification for cryo-EM analyses.** Viral stocks were propagated in Vero cells (ATCC) at ~90% confluency. MAYV was inoculated in monolayers at an approximate multiplicity of infection (MOI) of 0.01 and adsorbed for 1 h at 37 °C. Cell culture flasks were replenished with DMEM supplemented with 10% FBS and observed daily until ~80% of cytopathic effect took place. Infected cell culture supernatant was collected and precipitated in 8% PEG 8,000 overnight. MAYV-PEG precipitates were collected by centrifugation, resuspended in NTE buffer (NaCl 120 mM, Tris 20 mM, EDTA 1 mM) and centrifuged against a 22% sucrose cushion and later against a potassium tartrate gradient (10–35%). Purified MAYV bands were collected and concentrated using a 150 KDa cut-off filter (Amicon Ultra-2 100 Kd, Millipore). Final purified MAYV samples were stocked at −80 °C until use and quantified using plaque assays, to confirm stock viability and determine the yield of infective viral particles.

**Cryo-EM grid preparation.** Purified MAYV particles (5×10^10 PFU/mL) were sonicated (60 s pulse/30 s rest/60 s pulse) on an ultrasonic bath prior to grid preparation to ensure a homogeneous dispersion of viral particles. Using a Quantifoil R2/2 copper grid, 2.5 µL of sample were plunge-freezing on a Vitrobot Mark IV (Thermo Fisher Scientific) at 4 ºC and 100% humidity.

**Cryo-EM data collection.** All datasets were collected at LNNano Electron Microscopy Facility on a 300 kV Titan Krios G3i (Thermo Fisher) with Falcon 3EC direct electron detector (Thermo Fisher) using EPU software for automated data collection[43] to acquire four movies per hole. A total of 9100 movies in 4.26 s with 20 fractions (170 frames) per movie with a total dose (or accumulated dose) of ~30 e- Å^−2, with a dose of 1.5 e- Å^−2 per fraction in integration mode, at a nominal magnification of ×59,000, corresponding to a physical pixel size of 1.1 Å and a nominal defocus of −1.5 µm.

**Single particle analysis and image processing.** Cryo-EM data processing was performed largely following the Alignment-By-Classification protocol (ABC) as described in[44] using the Imagic-4d software system (version July 2020). A total of 175840 raw micrographs (4096 × 4096) (8792 movie stacks) were used in data processing and were firstly pre-processed through a posteriori camera correction to minimize spurious correlations from camera imperfections[45]. The normalized pixels values were multiplied by –1 to invert contrast and make proteins "white" and the background "dark", by standard convention. Then, micrographs were submitted to magnification anisotropy correction. An anisotropic magnification difference was found of around 1.5% was detected in a first round of eigen-vector analysis of amplitude spectra[47] of particle images was also corrected. The anisotropy corrected micrographs were submitted to a contrast transfer function (CTF) correction. A full-data-set CTF correction[46] was applied integrating the defocus information of the full date set (which ranges from −1.0 to −1.9 µm as calculated by CTF-Find function, as calculated by the Imagic program CTF-FIND), the amplitude spectrum of each image was determined and submitted to eigenvector

data-compression using the "modulation metric" and automatic unsupervised classification[47]. The resulting class averages were used to CTF parameters determination, which were passed to the individual images (movie frames). The CTF correction was applied by phase-flipping each image.

The CTF-corrected micrographs were resized in Fourier space to 1024 × 1024 dimensions and summed into movie stacks to be submitted to the particle picking, which was performed using a rotationally averaged particle from the own micrograph set. The particles were extracted from the original micrographs (4096 × 4096), band-pass filtered[44] (low frequency cut-off of 0.05 and a high frequency cut-off of 0.9) and each movie stack of particles were aligned and summed. A total of 79,000 initial particles were placed at a 800 × 800 box size with a circular soft mask of 0.77 radius (0.05 dropoff)[44] to remove information content outside the particle.

An initial three-dimensional reconstruction was obtained by angular reconstitution, which determined the relative Euler-angle orientations of the 2D-dimentional particle images, assuming the icosahedral symmetry. The first angular reconstitution cycle started from random assigned angles and projections of the first 3D reconstruction, which were used as an anchor set for a new cycle of angular reconstitution that was repeated until convergence. Then, the error associated to the angular reconstitution was used to filter the best image particles for further processing. The resulting 40,179 best particle images were then submitted to iterative processes of particle image rotation and translation alignment to 3D reprojections, followed by new cycles of angular reconstitution until achieving 3D resolution convergence. The final 3D map was band-pass filtered using a normalized amplitude spectrum (NAS) filter and masked using an automatic 3D mask to delineate the 3D map.

3D cryo-EM maps showing local (Fig. 1) and global resolution (Supplementary Fig. 1) were assessed in Imagic-4D by the Fourier shell correlation (FSC), using ½-bit threshold criterion[48–50]. For local resolution calculation, we used a kernel size of 30 pixels with interval between voxel samples of 5 pixels.

For density map visualization, we used UCSF Chimera version 1.13.1[51]. For clarity, global or cross-section density map views were built using a 2-sigma contour level and excluding densities blobs with size smaller than 15 A to remove noise. For views focused on residues of interest, we masked the density map within 3 A from the atoms and used 1.5–2.0 sigma contour levels. For glycan visualization, we zoned the density map within 10 A from all atoms (excluding noise densities blobs with size smaller than 6 A) and used a 2-sigma contour level.

**3D atomic structure modelling and refinement.** An initial 3D atomic model of MAYV IQT4235 strain was built by homology modelling using YASARA software[52]. One single unit of the CHIKV cryo-EM structure (PDB ID: 6NK5), composed by capsid and E1–E2 heterodimer from the asymmetric unit, was used as template to the homology modelling. The homology model coordinates were manually positioned into an asymmetric subunit of our MAYV cryo-EM map, regarding the T = 4 icosahedral organization, and then it was submitted to a rigid body fitting using *Fit in Map* function in Chimera software[51].

The resulting MAYV 3D coordinates were used as a starting point to a flexible refinement through the *phenix.real_space_refine* command-line module from the PHENIX software suite[53]. The refinement protocol was composed by rigid body fitting, local grid search, morphing, global minimization and simulated annealing steps. This refinement cycle was separately repeated for each capsid-E2-E1 unit forming the T = 4 lattice. Then, the four capsid-E2-E1 units were joined and submitted to a second refinement cycle. Finally, the quality of the final MAYV 3D atomic model was evaluated using MolProbity[54] in PHENIX GUI and Rotamer and Cβ outliers were manually inspected and corrected in Coot software[55]. A summary of MolProbity statistics is presented in Table S3.

**Prediction of N- and O-glycosylation.** N- and O-glycosylation sites in MAYV sequence were predicted using the NetNGlyc 1.0 and NetOGlyc 4.0 servers, respectively[56,57]. In NetNGlyc, in addition to canonical Asn-Xaa-Ser/Thr sequons, we performed the prediction of glycosylation on all ASN residues. Then, the predicted glycosylation sites were manually inspected for extra densities in the MAYV cryo-EM map.

**Protein extraction and digestion.** The viral proteins were obtained from purified viral stocks, separated in SDS-PAGE and the band containing E1 and E2 proteins was cut from the gel and submitted or not to de-N-glycosylation protocol. Fifteen micrograms of proteins were deglycosylated using 50 U Peptide-N-Glycosidase F (PNGase F, 500,000 U/mL, New England Biolabs) in 50 mM sodium phosphate buffer pH 7.5, 16 h at 37 °C, and other 15 μg of proteins were digested using EndoH (Endoglycosidase H, 500,000 U/mL, New England Biolabs). PNGase F or Endo H digested proteins and a non-digested protein sample (15 μg) were then reduced with dithiothreitol (10 mM for 30 min), alkylated with iodoacetamide (50 mM for 30 min at room temperature, in the dark) and digested with trypsin (1 μg) for 16 h at 37 °C. The peptides were extracted with 5% formic acid and 50% acetonitrile. The sample was dried in a vacuum concentrator and reconstituted in 10 μL of 0.1% formic acid.

**E1/E2 digestion for LC-MS/MS.** The peptide mixture (1.3 μL) was analysed using an LTQ Orbitrap Velos (Thermo Fisher Scientific) mass spectrometer coupled to nanoflow liquid chromatography on an EASY-nLC system (Proxeon Biosystems) with a Proxeon nanoelectrospray ion source. Peptides were subsequently separated in a 2–90% acetonitrile gradient in 0.1% formic acid using a PicoFrit analytical column (20 cm × ID75, 5 μm particle size, New Objective) at a flow rate of 300 nL/min over 65 min, in which a gradient of 35% acetonitrile is reached in 32 min. The nanoelectrospray voltage was set to 2.2 kV, and the source temperature was set to 275 °C. The instrument methods employed for LTQ Orbitrap Velos were set up in DDA mode. Full scan MS spectra (m/z 300–1600) were acquired in the Orbitrap analyzer after accumulation to a target value of 1e6. Resolution in the Orbitrap was set to r = 60,000, and the five most intense peptide ions (top 5) with charge states ≥2 were sequentially isolated to a target value of 50,000 and fragmented in higher-energy collisional dissociation (HCD) with a normalized collision energy of 40% with the resolution in the Orbitrap set to r = 7500 for MS/MS. Dynamic exclusion was enabled with an exclusion size list of 400 peptides, an exclusion duration of 60 s and a repetition count of 2. An activation time of 0.10 ms were used[58]. Complete lists of protein groups and peptides identified in each digestion condition are presented in Supplementary data files 1–6.

**MS data analysis.** Peak lists (msf) were generated from the raw data files using Proteome Discoverer version 1.4 (Thermo Fisher Scientific) with Sequest search engine and searched against The Human UniProt database (released January 2020, 98,510 sequences, and 38,155,965 residues) containing E1 and E2 Mayaro proteins sequences (2 sequences, 858 residues). Carbamidomethylation of cysteine (+57.021 Da) was considered a fixed modification and oxidation of methionine (+15.995 Da), deamidation (+0.984 Da), N-acetylglucosamine (+203.079 Da) and fucosylated N-acetylglucosamine (+349.333 Da) of asparagine as variable modifications, one trypsin missed cleavage and a tolerance of 10 ppm for precursor ions and 0.02 Da for fragment ions. A maximum of a 1% false discovery rate was set for both the protein and peptide identification. Possible N-Glycosylation sites were considered in deamidated asparagine, or asparagine with N-acetylglucosamine or fucosylated N-acetylglucosamine, in the motif N-X-S/T/C, with X ≠ P[59].

**N-glycan preparation for UPLC-MS/MS analysis.** For N-glycan analyses, the same de-N-glycosylation protocol with PNGase F mentioned above was applied, however in six batches of 15 μg of MAYV E1/E2 proteins, totalling 90 μg of glycoproteins. Glycans were collected in a supernatant phase of washing with 400 μL acetonitrile. A control sample containing the enzyme, digestion buffer and acetonitrile was also performed in parallel. Both samples were dried under vacuum at 40 °C and resuspended in 10 μl of H2O. These samples were incubated with 10 μl of labelling solution [DMSO and glacial acetic acid (70/30), 110 mg/ml of procainamide and 60 mg/ml sodium cyanoborohydrate[1] overnight at 65 °C, for reacting the released N-linked glycans with procainamide. After the incubation time, samples were dried under vacuum at 60 °C, resuspended in 10 μL of H2O and analyzed by UPLC-MS/MS.

**Analysis of N-glycans by UPLC-MS/MS.** Labelled glycans (7 μl) were analyzed using a 2.1 mm × 100 mm (1.7 μm, 130 Å) Acquity UPLC Glycan BEH Amide Column (Waters) on a Acquity H-Class Waters UPLC instrument (Waters, Milford, MA, USA) coupled with a hr-ESI-QqTOF Impact II (Bruker Daltonics, Billerica, MA, USA) mass spectrometer (MS Resolution 50,000). The chromatographic method was adapted from Kozak et al.[60], using 50 mM ammonium formate pH 4.4 buffer (A) and acetonitrile (B): 0–38.5 min, B: 76–58% (flow at 0.4 mL/min); 38.5–40.5 min, B: 58–40% (flow at 0.4 mL/min); 40.5–42.5 min, B: 40% (flow at 0.25 mL/min); 42.5–44.5 min, B: 40–76% (flow at 0.25 mL/min); 44.5–51.50 min, B: 76% (flow at 0.25 mL/min); 51.50–55 min, B: 76% (flow at 0.4 mL/min). For mass spectrometer parameters collected in positive mode: mass range was m/z 200–2500, spectrum rate of 1 Hz. Electrospray source was set up as: End Plate Offset: 500 V, Capillarity: 4500 V, Nebulizer: 4 Bar; Dry gas: 10 l/min; Dry temperature: 200 °C. In Tune, quadrupole ion energy: 5.0 eV; in collision cell: collision energy: 8 eV; pre pulse storage: 13 μs, using stepping tool: collision RF: 400–2000 Vpp; transfer time: 30–120 μs, timing: 80–20%; MS/MS only: collision energy: 100–250%; timing 50–50%. In MS/MS, scan mode: auto MS/MS; number of precursors: 3; threshold absolute 50 cts; CID spectra rate MS and MS/MS at 1 Hz. External calibrant used was sodium formate at 10 mM. Data were processed using the software Data Analysis 4.3 (Bruker).

**N-glycans identification from UPLC-MS/MS data.** N-glycans were located in the retention time range expected for this chromatographic method[60] Chromatographic peaks were manually selected and integrated. Each peak (#1 to #9) was inspected for extraction of (i) MS1 peaks, including a variety of doubled and tripled charged ions (2+, 3+) and adducts and (ii) MS2 spectra of the most intense peaks for each MS1. MS1 data allowed to predict the exact mass of all nine N-glycans. MS2 interpretation of seven peaks (#3 to #9) allowed to predict the composition of these N-glycans based on the fragmentation pattern, considering the losses or carbohydrate residues presented in the fragmentation spectra: m/z 162.05 ± 0.05 for hexoses and m/z 203.07 ± 0.05 for N-acetyl-hexosamines. For peaks #1 to #5, #8 and #9, MS2 presented the diagnostic peak m/z 441.27 ± 0.05, 1+, for GluNAc-procainamide ion; and MS2 for peaks #6 and #7 showed the diagnostic peak m/z 587.33 ± 0.05, 1+, for Fuc-GlucNAc-procainamide ion. All nine N-glycans

compositions were confirmed using the Glycoworkbench software software[61] and deposited at GlycoPOST (accession code GPST000171) following the MIRAGE guidelines[62].

**Pocket detection and characterization**. E1 and E2 ectodomains of MAYV, SINDV (PDB ID: 6IMM), EEEV (PDB ID: 6MX4), VEEV (PDB ID: 3J0C) and CHIKV (PDB ID: 6NK5) alphaviruses were used for cavity detection and characterization. First, a hydrophobic phospholipid tail (C18) was fitted to each structure and then, all ectodomains E1 and E2 were aligned. With these structures, we ran parKVFinder software[31], with Probe Out of 6.0 Å, Removal Distance of 0.6 Å and a space segmentation by an optimized box, to detect the cavity, around a reference ligand (with a 5.0 Å cut off except CHIKV with 8.0 Å cut off) and estimate its volume. We also determine the interface residues with side-chain contacts that compose the cavity, and evaluate the distribution of aliphatic apolar, aromatic, polar uncharged, negatively charged and positively charged residues. Furthermore, with these interface residues, we projected the Eisenberg consensus scale[63] of the residues that were closest to each surface point of the cavity. This procedure was repeated for MAYV C-proteins, except that we defined a Probe Out of 7.5 Å and a Removal Distance of 1.2 Å, and we did not limit the cavity around a reference ligand.

**Reporting summary**. Further information on research design is available in the Nature Research Reporting Summary linked to this article.

## Data availability

The cryoEM datasets generated and analysed during the current study are available in the EMDB repository, entry EMD-22961, and in the PDB repository, entry 7KO8. The mass spectrometry-based proteomics data have been deposited to the ProteomeXchange Consortium via the PRIDE partner repository[64] with the dataset identifier PXD024432. The UPLC-MS/MS datasets are deposited at GlycoPOST under the accession code GPST000171.

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

## Acknowledgements

We thank Mr. Valber Ferreira for his technical assistance. We also thank Dr. Marjorie Bruder for reviewing our manuscript. We thank LNNano/CNPEM for the use of electron microscopy facility (TEM-23519, TEM-24308) and for supporting part of computational analyses, and LNBR/CNPEM for providing the oligomannose standards. This study was supported by grants from FAPESP (2018/03917-6, 2017/15340-2, 2018/00629-0), CNPq (440379/2016-4), CAPES (130767/2016-01) and Serrapilheira Institute, grant number Serra-1709-19681.

## Author contributions

L.D.C., A.C., R.P.F.R., R.F., C.M.C. and L.L. performed experiments. H.V.R.F., J.V.S.G., R.F., A.C.P., A.F.P.L., R.V.P. and P.S.L.O. were involved in data processing. H.V.R.F., L.D.C., R.P.F.R., R.F., A.F.P.L., D.B.B.T., P.S.L.O. and R.E.M. analysed and interpreted data. H.V.R.F., L.D.C., D.B.B.T., P.S.L.O. and R.E.M. wrote the manuscript. R.V.P., P.S.L.O. and R.E.M. conceived the study.

## Competing interests

The authors declare no competing interests.
