## [Peer Review File · Nature Communications]

REVIEWER COMMENTS

Reviewer #1 (Remarks to the Author):

Cryo-EM structure of the mature and infective Mayaro virus at 4.4 Å resolution reveals new features of arthritogenic alphaviruses

Filho et al

In this manuscript, the authors have used single particle cryoEM to perform a reconstruction of MAYV to an average resolution of 4.4Å. The authors have analyzed this structure in comparison to, mainly, the cryoEM structure of CHIKV.

The main finding in the investigations seems to be a molecular handshake between the viral envelope proteins involving glycosylation. The biological relevance of this interaction is unclear although it is certainly intriguing. The data in the rest of the manuscript is unfortunately mostly a repetition of previously published alphavirus structures. As the authors describe, there are already several published alphavirus structures in the reported resolution range, which provide a rather comprehensive database of structural details. Therefore, this manuscript may not be suitable for Nature Communications in its present state.

Here are some comments that might help the authors.

Line 75: Figure S10 should not be cited so early in the text before other supplementary figures. Please renumber the figures as they appear in the text.

Lines 76-79: This sentence is misleading. It is not true that most alphavirus structures have been determined using VLP samples. On the contrary, there is ample structural data on purified virus particles such as Semliki Forest virus (Fuller Lab), Ross River virus (Baker Lab), Sindbis virus (Rossmann Lab; Zhang Lab), Chikungunya virus (Rossmann Lab), WEEV (Weaver Lab), VEEV (Chiu Lab), and EEEV (Rossmann Lab). Please remove this sentence.

Figure S1C: This is not a good micrograph. The contrast is so poor that the particles are not easily visible. Either low-pass filter and bin this image to improve the contrast or use a different micrograph.

Figure 1B: It seems that the white triangle represents the asymmetric unit. This is incorrectly drawn. The $i3$ axis must not be at the center of the asymmetric unit. Instead, $i3$ axes should be at two vertices of the asymmetric unit. The authors should look up any of the papers showing this asymmetric unit, such as the Cell Reports paper on the EEEV structure.

Lines 114-116: An alternative explanation for the low resolution is simply propagation of error in the orientation angle. As one moves radially from the center of the particle to the surface, the linear displacement associated with the angle grows larger. As a consequence, the quality of the map could drop from the center to the outside.

Lines 125-126: This statement about "well evidenced" is only partly true. For instance, in Fig 2C, panel 3, the side chains are shown as sticks in an iso-contour surface that shows no discernable side-chain features. This is to be expected at the reported resolution.

Lines 173-184: Why did the authors need to determine a structure to arrive at these conclusions? All these inferences could have been made using the protein sequence, except for the glycosylation. Relying on a structure to observe glycosylation is not a good strategy

anyway. An enzymatic digestion of sugars or mass spectrometry would provide more reliable results about glycosylation.

General comment: CryoEM maps do not show density but rather potential. The authors could replace the word density with a more appropriate term, such as "feature".

CryoEM data collection table: The defocus range is missing.

Reviewer #2 (Remarks to the Author):

Cryo-EM structure of the mature and infective Mayaro virus at 4.4 Å resolution being reported by Filho et al., in the manuscript reveals new features of arthritogenic alphaviruses. Authors of this paper have described detailed interactions between MAYV envelope proteins majorly hydrophobic pocket formed between E1 and E2 spike proteins and their conformational epitopes. They have also given insights on MAYV glycosylation residues in E1 and E2 that may affect MXRA8 host receptor binding, and a molecular "handshake" between MAYV spikes formed by N262 glycosylation in adjacent E2 proteins. The information that have been provided in the paper are unique, and might lead to better understanding of antibody interaction with envelope proteins on the surface of alphaviruses in general. The manuscript have been drafted well however we have few concerns which are mentioned below

1. Here in this paper by Powell et al., 2020 they have used an average resolution of each map calculated at 0.143 from FSC curves was 6.3Å° for RRV, 5.3Å° for CHIKV, 5.3Å° for MAYV, and 4.8 Å° for native MAYV. However, in your paper a FSC of 0.5 have been used for structure prediction. Please elaborate the criteria for selection of this FSC cut-off value.
2. The authors have reported the presence of N and O glycosylation sites on E1 and E2, through CryoEM data, we would like to know if there has been other biochemical or biophysical experiments done to validate presence of these sites on viral surface for instance in this paper Watanabe, Yasunori, Joel D. Allen, Daniel Wrapp, Jason S. McLellan, and Max Crispin. "Site-specific glycan analysis of the SARS-CoV-2 spike." *Science* (2020)., the researchers have validated the glycosylation sites on spike protein of SARS CoV2 by LC-MS as well.

Response to reviewers

Cryo-EM structure of the mature and infective Mayaro virus at 4.4 Å resolution reveals new features of arthritogenic alphaviruses

Helder R. Filho, Lais D. Coimbra, et al. 2021

Thank you for critically assessing our manuscript. Please find enclosed the reviewed version of our manuscript and a point-by-point reply to each comment. We were able to perform all experiments requested, including the characterization of MAYV glycans using mass spectrometry. All changes to the manuscript files were marked in **red**.

We hope our manuscript is now suitable for publication in Nature Communications.

Reviewer#1:

In this manuscript, the authors have used single particle cryoEM to perform a reconstruction of MAYV to an average resolution of 4.4Å. The authors have analyzed this structure in comparison to, mainly, the cryoEM structure of CHIKV. The main finding in the investigations seems to be a molecular handshake between the viral envelope proteins involving glycosylation. The biological relevance of this interaction is unclear although it is certainly intriguing.

R: Indeed, we show a novel structural feature of MAYV whose function is yet to be determined. We wish to further investigate this feature soon.

The data in the rest of the manuscript is unfortunately mostly a repetition of previously published alphavirus structures. As the authors describe, there are already several published alphavirus structures in the reported resolution range, which provide a rather comprehensive database of structural details. Therefore, this manuscript may not be suitable for Nature Communications in its present state.

R: Alphaviruses are closely related and expected to share basic structural features. However, we lack a better understanding of alphavirus biology that has not been solved by inference from CHIKV or SINV. Using MAYV as a model organism, we confirmed the structural similarities to other alphaviruses, while pointing out relevant differences and features never observed before. Examples include the novel handshake configuration resulting of N262 glycosylation, the in-depth analysis of the E1-E2 and C hydrophobic pockets, the spatially conserved N141 glycosylation and a structural insight on why alphaviruses bind to MXRA8 with different affinities. Therefore, our findings go beyond the MAYV structure *per se*, and should be suitable for Nature Communications due to novelty and quality.

Here are some comments that might help the authors. Line 75: Figure S10 should not be cited so early in the text before other supplementary figures. Please renumber the figures as they appear in the text.

R: We reorganized the order of supplementary figures in the files.

Lines 76-79: This sentence is misleading. It is not true that most alphavirus structures have been determined using VLP samples. On the contrary, there is ample structural data on purified virus particles such as Semliki Forest virus (Fuller Lab), Ross River virus (Baker Lab), Sindbis virus (Rossmann Lab; Zhang Lab), Chikungunya virus (Rossmann Lab), WEEV (Weaver Lab), VEEV (Chiu Lab), and EEEV (Rossmann Lab). Please remove this sentence.

R: We removed the misleading sentence in lines 75-78.

Figure S1C: This is not a good micrograph. The contrast is so poor that the particles are not easily visible. Either low-pass filter and bin this image to improve the contrast or use a different micrograph.

R: We provided a better micrograph for Supplementary Figure 2C (previously S1C). The micrograph was processed by applying a band-pass filter in Imagic-4D software to suppress very low-frequency information (less than 0.02 of Nyquist frequency), removing ramps of background fluctuations, and to suppress very high-frequency information (higher than 0.4 of Nyquist frequency), allowing a better visualization of low frequency content. After filtering, we resized the micrograph to 512x512 dimensions.

Figure 1B: It seems that the white triangle represents the asymmetric unit. This is incorrectly drawn. The $i3$ axis must not be at the center of the asymmetric unit. Instead, $i3$ axes should be at two vertices of the asymmetric unit. The authors should look up any of the papers showing this asymmetric unit, such as the Cell Reports paper on the EEEV structure.

R: The white dashed triangle represented the icosahedron face, not the asymmetric unit, as in Figure 1A. We agree with Reviewer#1 that a representation of the asymmetric unit is more useful and a common practice in Cryo-EM papers. We removed the icosahedral cage from Figure 1A and the previous white dashed triangle from Figure 1B and included the white dashed triangle representing the asymmetric unit, with vertices at $i3$ and 5-fold axes. In addition, Figure 1E was replaced by a zoomed view of the E1-E2-C asymmetric unit.

Lines 114-116: An alternative explanation for the low resolution is simply propagation of error in the orientation angle. As one moves radially from the center of the particle to the surface, the linear displacement associated with the angle grows larger. As a consequence, the quality of the map could drop from the center to the outside.

R: We agree with Reviewer#1 that propagation of error may contribute to the lower resolution in distal parts of the virus. However, whether that was the only reason for the lower resolution, we would expect a gradient local resolution, from higher (at the center) to lower (at the surface) resolutions values. On the contrary, we can observe some regions of higher resolution than the most external portion of E2 protein at a similar distance to the center of the particle. In addition, we also observe lower resolution regions at more internal regions, such as the capsid and transmembrane regions. Even so, to minimize some degree of error propagation contribution in data interpretation, we restricted local resolution comparison of the most external portion of E2 to only E1-E2 ectodomains, which are in a similar radial distance from the center. We reformulated lines 103-105 in the revised manuscript. The next sentence (lines 105-107) remains unchanged since a comparison is already made between regions at the same radial distance from the center.

Lines 125-126: This statement about "well evidenced" is only partly true. For instance, in Fig 2C,

panel 3, the side chains are shown as sticks in an iso-contour surface that shows no discernable side-chain features. This is to be expected at the reported resolution.

R: We removed the statement in lines 113-114 and replaced by: “Newly **putative** side chain contacts are **evidenced** by our cryo-EM map.”

Lines 173-184: Why did the authors need to determine a structure to arrive at these conclusions? All these inferences could have been made using the protein sequence, except for the glycosylation.

R: Our purpose in this section was to point out differences between MAYV and CHIKV that may be relevant for serological cross-reactivity. We indeed start from the protein sequence level to identify amino acid deletions or substitutions on MAYV-CHIKV alignment but move on to the actual impact of these changes in E1 and E2 3D protein structures. For example, there are 2 sequential missing residues in MAYV E1 in comparison to CHIKV. These missing residues do not disrupt the overall tertiary E1 III subdomain structure but result in a shorter solvent-exposed beta-strand, as observed in Figure 3A. Consequently, contacts between two antiparallel beta-strands reduce in number, increasing the distance between beta-strands that could imply a mobility change. All these structural findings cannot be accessed solely from a protein sequence and rather depend on interpretation of the 3D protein structure. The availability of 3D structural information, especially from this region, can be a target for the rational design of antibodies to differentiate MAYV from CHIKV. To clarify our purpose in this section (“Comparison of MAYV and CHIKV ectodomains and neutralizing antibody sites”) and to better correlate MAYV 3D structure to our analysis, we modified sentences in the first paragraph (lines 154-167).

Relying on a structure to observe glycosylation is not a good strategy anyway. An enzymatic digestion of sugars or mass spectrometry would provide more reliable results about glycosylation.

R: We agree with Reviewer that further characterization of glycosylations is necessary. We then performed LC-MS/MS analysis of E1 and E2 proteins using three strategies: (i) only trypsin digestion, (ii) complete release of N-glycans through PNGase F digestion followed by trypsin digestion, or (iii) Endo H digestion following trypsin digestion. The results indicate N141 in E1 and N262 in E2 are glycosylated. We also investigated the composition of sugars, represented by combinations of N-acetyl-hexosamines (HexNAc) and hexoses (Hex). Our data corroborates the glycan chains (initial monomers) observed and interpreted from the MAYV cryoEM map and the LC-MS/MS analysis of E1/E2 protein digests with PNGase F and EndoH mentioned above. Altogether, considering the canonical glycosylation residues expected in mammalian cells, N-acetylglucosamine (NAG) and mannose (MAN) can be suggested as MAYV glycans. Data are described in greater detail in lines 223-262 and are represented in Figure 4D, Supplementary Figure 11 and 12, Supplementary table 5 and Supplementary File 1. UPLC-MS/MS raw data and data interpretation of the released N-glycans derivatized with procainamide can be accessed at GlicoPOST under ID: GPST000171 (Reviewer Login details: <https://glycopost.glycosmos.org/>)

preview/127558071760406e2364a2e, pin Code: 7622). After manuscript acceptance the link will be made public and further deposited at UniCarb-DB.

General comment: CryoEM maps do not show density but rather potential. The authors could replace the word density with a more appropriate term, such as "feature".

R: We use terms such as “cryo-EM density map” or “density” in accordance to recent publications in the field of Cryo-EM: “**Cryo-EM structure of eastern equine encephalitis virus in complex with heparan sulfate analogues**” - Michael Rossmann’s Group, PNAS – 2020, “**Cryo-EM structure of a 40 kDa SAM-IV riboswitch RNA at 3.7 Å resolution**” - Wah Chiu’s Group, Nature Communications – 2019, “**Improvement of cryo-EM maps by density modification**” -, Pavel V. Afonine and Thomas C. Terwilliger Groups, Nature Methods – 2020, “**Information: to Harvest, to Have and to Hold**” - Marin Van Heel’s Group, Arxiv – 2020, “**Single-particle cryo-EM at atomic resolution**” - Sjors Scheres, Nature – 2020, and many others. We understand the use of certain terms in cryo-EM may change in time but have decided to keep using “density” for clarity and uniformity.

CryoEM data collection table: The defocus range is missing.

R: The defocus information was presented in a table named “Data collection and processing” in the Methods section. Based on the Reviewer#1 comment, we removed the table from Methods and included the required information in Supplementary Table 2 “Cryo-EM data collection and processing”, to improve data presentation. We also included the estimated defocus values of micrographs using Imagic CTF-FInd function under “Single Particle Analysis and Image Processing” in Methods.

Reviewer#2:

Cryo-EM structure of the mature and infective Mayaro virus at 4.4 Å resolution being reported by Filho et al., in the manuscript reveals new features of arthritogenic alphaviruses. Authors of this paper have described detailed interactions between MAYV envelope proteins majorly hydrophobic pocket formed between E1 and E2 spike proteins and their conformational epitopes. They have also given insights on MAYV glycosylation residues in E1 and E2 that may affect MXRA8 host receptor binding, and a molecular “handshake” between MAYV spikes formed by N262 glycosylation in adjacent E2 proteins. The information that have been provided in the paper are unique, and might lead to better understanding of antibody interaction with envelope proteins on the surface of alphaviruses in general. The manuscript have been drafted well however we have few concerns which are mentioned below

R: We appreciate the positive comments from Reviewer #2. We hope to address your concerns adequately.

1. Here in this paper by Powell et al., 2020 they have used an average resolution of each map calculated at 0.143 from FSC curves was 6.3Å° for RRV, 5.3Å° for CHIKV, 5.3Å° for MAYV, and 4.8 Å° for native MAYV. However, in your paper a FSC of 0.5 have been used for structure prediction. Please elaborate the criteria for selection of this FSC cut-off value.

R: We agree with reviewer #2 in the concerning of using 0.5 cut-off value. In the current version of the manuscript, local resolution was estimated using Imagic-4D based on the ½ -bit criteria (Van Heel & Schatz, 2020, DOI: arXiv:2009.03223; Van Heel & Schatz, 2017 DOI: <https://doi.org/10.1101/224402>, Van Heel & Schatz 2005, DOI: <https://doi.org/10.1016/j.jsb.2005.05.009>). Cryo-EM maps from Figure 1 and 8 were updated. The average resolution was already estimated based on the same criteria (Supplementary Figure 1).

2. The authors have reported the presence of N and O glycosylation sites on E1 and E2, through CryoEM data, we would like to know if there has been other biochemical or biophysical experiments done to validate presence of these sites on viral surface for instance in this paper Watanabe, Yasunori, Joel D. Allen, Daniel Wrapp, Jason S. McLellan, and Max Crispin. "Site-specific glycan analysis of the SARS-CoV-2 spike." Science (2020)., the researchers have validated the glycosylation sites on spike protein of SARS CoV2 by LC-MS as well.

R: We performed LC-MS/MS analysis of E1 and E2 proteins for glycosylation site characterization using three strategies: (i) only trypsin digestion, (ii) complete release of N-glycans through PNGase F digestion followed by trypsin digestion, or (iii) Endo H digestion following trypsin digestion. The results indicate N141 in E1 and N262 in E2 are glycosylated. Please find a detailed description of our findings in lines 223-245. Data are shown in Supplementary Figure 11 and Supplementary File 1. Raw data can be accessed via the ProteomeXchange Consortium via the PRIDE partner repository, where a Reviewer account has been set up (Username: reviewer_pxd024432@ebi.ac.uk, password: JMziJh10).

REVIEWER COMMENTS

Reviewer #2 (Remarks to the Author):

The authors have addressed the queries very well.

MAYV is expected to have structural similarities to other alphaviruses. However, authors have analysed the data carefully and have pointed out relevant differences and features not observed before in other alphavirus cryoEM structures such as handshake configuration and detail

analysis of the glycoprotein and C hydrophobic pocket etc

Therefore, I think the manuscript is in good shape and appropriate for NCOMMS.